# TLQP-21, A VGF-Derived Peptide Endowed of Endocrine and Extraendocrine Properties: Focus on In Vitro Calcium Signaling

**DOI:** 10.3390/ijms21010130

**Published:** 2019-12-24

**Authors:** Elena Bresciani, Roberta Possenti, Silvia Coco, Laura Rizzi, Ramona Meanti, Laura Molteni, Vittorio Locatelli, Antonio Torsello

**Affiliations:** 1School of Medicine and Surgery, University of Milano-Bicocca, 20900 Monza, Italy; silvia.coco@unimib.it (S.C.); laura.rizzi@unimib.it (L.R.); r.meanti@campus.unimib.it (R.M.); laura.molteni@unimib.it (L.M.); vittorio.locatelli@unimib.it (V.L.); antonio.torsello@unimib.it (A.T.); 2Department of Systems Medicine, University of Roma Tor Vergata, 00133 Roma, Italy; roberta.possenti@gmail.com

**Keywords:** VGF, TLQP-21, calcium (Ca^2+^), endocrine, extraendocrine, complement C3a receptor-1 (C3aR1), stromal interaction molecules (STIM), calcium release-activated calcium channel (CRAC)-Orai1, transient receptor potential channel (TRPC), KCa3.1 current

## Abstract

VGF gene encodes for a neuropeptide precursor of 68 kDa composed by 615 (human) and 617 (rat, mice) residues, expressed prevalently in the central nervous system (CNS), but also in the peripheral nervous system (PNS) and in various endocrine cells. This precursor undergoes proteolytic cleavage, generating a family of peptides different in length and biological activity. Among them, TLQP-21, a peptide of 21 amino acids, has been widely investigated for its relevant endocrine and extraendocrine activities. The complement complement C3a receptor-1 (C3aR1) has been suggested as the TLQP-21 receptor and, in different cell lines, its activation by TLQP-21 induces an increase of intracellular Ca^2+^. This effect relies both on Ca^2+^ release from the endoplasmic reticulum (ER) and extracellular Ca^2+^ entry. The latter depends on stromal interaction molecules (STIM)-Orai1 interaction or transient receptor potential channel (TRPC) involvement. After Ca^2+^ entry, the activation of outward K^+^-Ca^2+^-dependent currents, mainly the K_Ca3.1_ currents, provides a membrane polarizing influence which offset the depolarizing action of Ca^2+^ elevation and indirectly maintains the driving force for optimal Ca^2+^ increase in the cytosol. In this review, we address the main endocrine and extraendocrine actions displayed by TLQP-21, highlighting recent findings on its mechanism of action and its potential in different pathological conditions.

## 1. Introduction 

Calcium (Ca^2+^) is a universal signaling molecule impacting nearly every aspect of cellular life by regulating a variety of cellular processes such as fertilization, gene transcription, cell proliferation, programmed cell death, neurotransmission, muscle contraction, and cell signaling [1]. The Ca^2+^ ion could act as second messenger released inside the cells following the interaction of different endogenous molecules with their plasma membrane (PM) receptors. However, it can also enter directly into the cells from the extracellular space to deliver information, without the intermediation of other first or second messengers. The most distinctive feature of Ca^2+^ is its Janus-faced action: while it is crucial for the homeostasis and the correct functioning of the cells, an uncontrolled excess of its influx or release from intracellular stores can turn it into a toxic agent leading to cell apoptosis or necrosis [1]. Thus, the maintenance of adequate Ca^2+^ concentrations across extracellular and intracellular compartments of the cells is essential for cellular homeostasis. At resting state, free intracellular Ca^2+^ levels are strictly kept low (10–100 nM), in comparison to the endoplasmic/sarcoplasmic reticulum (ER/SR) levels (60–500 µM) [2] and the extracellular medium (1.8 mM) [3]. Ca^2+^ signals are generated within a wide spatial and temporal range through a huge variety of Ca^2+^ channels as well as pumps and exchangers located in the PM and/or ER/SR. The increase of the intracellular Ca^2+^ concentration can result from the activation of cation channels at the PM, like: (i) Transient receptor potential channels (TRPC), a large group of non-selective cation channels activated by a several stimuli and mostly permeable to Ca^2+^; (ii) Voltage-gated ion channels (VGCs), activated by changes in the electrical PM potential near the channel proteins or mechanosensitive channels; (iii) ER/SR-associated inositol-3-phosphate receptor (IP3R) and ryanodine-receptor (RyR) channels; and iv) Ca^2+^ release activated Ca^2+^ (CRAC) channels composed of two cellular proteins, Ca^2+^-sensing stromal interaction molecule 1 (STIM1) and pore-forming, Orai1. The depletion of intracellular Ca^2+^ stores is sensed by STIM which interact with CRAC-Orai1, triggering store-operated Ca^2+^ entry (SOCE) currents, exploited to regulate basal Ca^2+^, to refill intracellular Ca^2+^ stores, and to execute a wide range of specialized activities. On the other side, the systems for Ca^2+^ extrusion consist of PM exchangers, as well as the low-affinity high-capacity sodium-calcium exchanger (NCX), the high-affinity low-capacity Plasma Membrane Calcium ATPase (PMCA), and the sarco-endoplasmic reticulum calcium ATPase (SERCA) pump, designed for Ca^2+^ sequestration in the ER/SR [4]. Binding to PM receptors, a large number of endogenous molecules can induce a rapid cytosolic Ca^2+^ surge, which in turn activates a variety of signaling pathways. 

VGF is a large neuropeptide precursor that generates a family of peptides endowed with different biological activities [5]. Among them, TLQP-21 has attracted the interest of several research groups by virtue of its endocrine and extraendocrine effects. Recently, in the attempt to elucidate TLQP-21 mechanism of action, researches identified a putative TLQP-21 receptor, evidencing that most of its actions are mediated by the second messenger Ca^2+^. This review aims to summarize the recent advances in TLQP-21 investigations. We then provide an integrated view on the recent reports supporting the role of Ca^2+^ in mediating TLQP-21 mechanism of action.

## 2. VGF and VGF-Derived Peptides 

VGF was discovered in 1985 in the framework of the studies designed to study neurotrophins and growth factors role in the development, maintenance, and normal functioning of the peripheral and central nervous systems (PNS and CNS) [5,6]. VGF (non-acronymic name) was initially identified as a nerve growth factor (NGF)-regulated transcript in rat PC12 pheochromocytoma cells, on the basis of its rapid induction in PC12 cells treated with NGF. The name VGF was chosen based on the selection of a clone from plate “V” of a NGF-induced PC12 cell cDNA library and should not be confused with VEGF (vascular endothelial growth factor) [5]. Further studies have demonstrated that VGF is upregulated also by other neurotrophins, including brain-derived neurotrophic factor (BDNF) and neurotrophin-3 (NT-3), in neuronal targets such as cortical or hippocampal neurons [5,6,7]. However, VGF mRNA levels are only marginally increased by epidermal growth factor (EGF), fibroblast growth factor (FGF), interleukin-6 (IL-6), and insulin, despite the capacity of these proteins to induce transcription of other immediate early genes in the PC12 cell line [8].

As above mentioned, VGF gene encodes for a neuropeptide precursor of 68 kDa with a tissue-specific pattern of expression prevalently in the CNS, but also in the PNS and in various endocrine cells [9]. In the CNS, VGF immunoreactivity is most abundant in the olfactory system, cerebral cortex, hypothalamus, hippocampus, in thalamic, septal, amygdaloid and brainstem nuclei, and spinal cord neurons; in the PNS, high levels of VGF are expressed in the primary sensory neurons and in neurons of sympathetic ganglia [9,10,11,12,13]. 

Different populations of endocrine cells in the adenohypophysis, adrenal medulla, gastrointestinal tract, and pancreas express VGF [9,10,11,12,13]. The studies on VGF regulation and distribution suggest that this molecule plays an important role in the developing and/or adult nervous and neuroendocrine systems, on feeding and energy expenditure, locomotor activity, sympathetic nerve activity, gastrointestinal motility and/or secretion, and pancreatic hormone secretion. An involvement in the regulation of reproduction and fertility is also likely, in view of the its pituitary localization and modulation with the estrous cycle [11,12]. The precursor VGF protein is composed by 615 (human) and 617 (rat, mice) amino acids [14]. One of the feature of the VGF protein structure is the presence of specific sequences rich in basic amino acid residues that represent potential cleavage sites for protein convertases of the kexin/subtilisin-like serine proteinase family, a class of enzymes that activate other proteins. Upon VGF cleavage by prohormone convertase (PC) 1/3 and PC2, a variety of bioactive peptides of low molecular weight are generated [13]; they are stored in dense core secretory vesicles and secreted upon stimuli [15]. More than 10 different VGF-derived peptides were detected in rat brain, bovine pituitary and human cerebrospinal fluid (CSF) [16,17,18]. Among them, those endowed with specific neuronal activities are: (i) neuroendocrine regulatory peptides-1 and -2 (NERP-1 and NERP-2); (ii) NAPP-129, also called VGF 20, based on its molecular weight (20 kDa); (iii) TLQP-62 (also known as VGF 10, 10 kDa) and TLQP-21; (iv) HHPD-41; (v) AQEE-30 (or Peptide V) and AQEE-11; vi) and LQEQ-19. Figure 1 summarizes the schematic representation of VGF-derived peptides and Table 1 indicates their activities. 

### 2.1. TLQP-21: A VGF-Derived Peptide 

By convention, the acronym TLQP designates the 4 N-terminal amino acid sequence of the peptides, i.e., Thr-Leu-Gln-Pro, resulting from the cleavage on the VGF precursor, at the specific R-P-R (Arg-Pro-Arg) processing site found at rat/mouse VGF_553–555_ (VGF_551-553_ in human), and variably extended to the VGF precursor C-terminus. The number is the amount of amino acid residues composing the peptide [5]. The TLQP family encompasses two major peptides, TLQP-62 and TLQP-21. Compared to other VGF-derived peptides, TLQP peptides expression is low in the brain, mainly restricted to the hypothalamus, in a subpopulation of neurons projecting to a discrete areas of median eminence, in the prefrontal cortex, and in the limbic region [18], whilst are more represented in hypothalamic–pituitary axis, plasma, as well as in several peripheral locations [19,20]. The tissue distribution of TLQP-21 was recently explored by studies performed with a radiolabeled fluorine-18 [18F] short analog of TLQP-21, administered intravenously (i.v.) in mice and analyzed by PET imaging [21]. Consistently with previous observations [22,23], high uptake of [18F]-JMV5763 was found peripherally, in stomach, intestine, kidney, liver, adrenal gland and lungs, whereas in the CNS the uptake was extremely low. In vivo PET analysis was consistent with the findings obtained from ex vivo experiments [21].

In recent years, TLQP-21 has been widely investigated for its biological effects and functions. Studies in this field were prompted by the observation that VGF knockout mice (KO) are lean, hypermetabolic, hyperactive, and possess an obesity resistant phenotype, characterized by markedly reduced leptin levels and fat stores [10]. These features, coupled with altered hypothalamic expression of factors involved in eating behavior, such as proopiomelanocortin (POMC), neuropeptide Y (NPY), and agouti-related peptide (AGRP), suggested that VGF could play a key role in the control of energy homeostasis [10].

### 2.2. TLQP-21: Endocrine Activities 

TLQP-21 appears to be a multifunctional peptide involved in various endocrine functions. Depending from the source of its release (neurons or endocrine cells), it is indeed difficult to establish a clear endocrine effect or a paracrine or autocrine action. Despite the half-life after a single i.v. injection is initially of 0.97 min, the complete terminal value is about 110 min due to tissue uptake and renal clearance [24]. Parenteral or intra-cerebro-ventricular (i.c.v.) injection clearly results in a strong metabolic effect. This aspect will be discussed in the Section 2.3. Various models of biological activity have been proposed.

#### 2.2.1. Lactogenic Effect 

In GH3 cells, a somatomammotropic cell line that synthesize growth hormone (GH) and prolactin (PRL), TLQP-21 induces differentiation toward a more lactogenic phenotype, increasing PRL and decreasing GH production [25]. This is consistent with immunohistochemistry data that show high level of VGF C-terminal peptides expression in pituitary lactotropic cells of seasonal reproductive sheep [26]. In GH3 cells, TLQP-21 stimulates phosphorylation of different kinases and Ca^2+^ mobilization from thapsigargine (TP) sensitive stores (see Section 2.5). There are no evidences of in vivo lactogenic studies, but chronic i.c.v. TLQP-21 delivery consistently failed to stimulate the GH/IGF-1 axis [27].

#### 2.2.2. Effect on the Reproductive Tract 

VGF immunofluorescence was detected in gonadotropic pituitary cells where it was modulated during estrous cycle [28]. Two different VGF derived peptides could be involved in reproductive functions: NERP1 (PESA-25), which inhibits Follicle-Stimulating Hormone (FSH) expression [29], and TLQP-21 that stimulates gonadotropin production [30,31]. In prepubertal male rat, TLQP-21 promotes directly at the pituitary level the secretion of Luteinizing Hormone (LH), whereas in vitro it enhances the testosterone production induced by human chorionic gonadotropin (hCG) in testicular tissue from adult rats [31]. In female rats, systemic administration of TLQP-21 boosts LH and FSH secretion and ovarian maturation depending on pubertal and nutritional stage. In adult female, central administration of TLQP-21 differently affects LH secretion depending on the estrous cycle phase [30].

#### 2.2.3. Effect on Endocrine Pancreas 

VGF and its derivatives are highly expressed in pancreatic islet β-cells where VGF plays a critical role on biogenesis of the secretory granules, increasing insulin cargo transport [32,33]. A paracrine and autocrine effects are probably involved in its neuroendocrine action. The role of TLQP-21 on pancreatic function has yet to be clarified. Although it was firstly postulated capable to slow diabetes development via the enhancement of islet β-cell survival and function [34], in vitro stimulation of isolated perfused rat pancreas failed to enhance insulin, somatostatin, or glucagone release [35]. These results have been obtained with physiological amounts of peptide ranging up to 1 micromolar, although it was also observed that higher concentrations of TLQP-21 (up to 10 micromolar) slightly enhanced insulin release, by increasing calcium influx in INS-1 cells (Possenti unpublished data). 

#### 2.2.4. Effect on the Gastrointestinal Tract 

TLQP-21 is highly expressed in enterochromaffin and neuronal cells of the stomach [20]. One of the first study on TLQP-21 showed that this peptide is involved in gastric motor function both in vitro and in vivo [36]. In rat longitudinal strips (with less potency in other GI tracts, Possenti unpublished data) TLQP-21 stimulates muscle contraction, whereas central TLQP-21 administration significantly decreases gastric empting. Moreover, i.c.v. TLQP-21 administration exerts a protective effect on gastric mucosa exposed to ethanol toxicity [37] and significantly reduces gastric acid secretion [38]. No such effects appeared after single or chronic peripheral administration, indicating the involvement of sensory and autonomic nervous fiber.

On pancreatic lobules TLQP-21 induced ex vivo an increase of pancreatic amylase release, but failed to stimulate directly the acinar cells, suggesting an indirect mechanism of action [39].

### 2.3. TLQP-21 Metabolic Actions 

TLQP-21 showed a specific role in nutrition and metabolism [16]. Chronic i.c.v. TLQP-21 administration by osmotic minipumps in mice fed with a standard diet increases energy expenditure (EE), affects adrenergic function, and lipid profile, lowering triglycerides (TG), without altering body weight and food intake. Therefore, it seems that central infusion of TLQP-21 modify energy balance without determining a shift in energy homeostasis. This effect has been in part related to a mechanism involving CNS cyclooxygenase-2 (COX-2) and therefore the prostaglandin (PG) production, which could be responsible of the increased body temperature and EE [16]. The lack of changes in the major anorexigenic or orexigenic neuropeptide, such as AGRP, NPY, melanin-concentrating hormone (MCH), POMC, and corticotropin-releasing hormone (CRH), supports the idea that the hypothalamus does not mediate TLQP-21 effects, which is more in agreement with the activation of brainstem nuclei, downstream of the hypothalamus [16]. The increased expression of β_3_-adrenergic receptor (AR), peroxisome proliferator activated receptor (PPAR)-δ and uncoupling protein (UCP)-1 in the white adipose tissue (WAT) suggested an additional peripheral catabolic action [16]. Indeed, in high-fat diet-induced obese mice, TLQP-21 central administration counteracts the early phase of obesity, preventing the weight and adiposity gain, and chronic subcutaneous (s.c.) treatment increases lipolysis in murine adipocytes. [22,40]. The lipolytic actions of TLQP-21 could be attributable to a mechanism involving the norepinephrine (NE)/β-AR activation. This idea is corroborated by the presence of VGF peptide in the axonal terminations of sympathetic fibers in WAT and by the demonstration that TLQP-21 binding to murine adipocytes causes a reduction of the diameter of the cells following the activation of the sympathetic system [22]. By interacting with its receptor expressed on adipocyte membrane, TLQP-21 could potentiate the β-AR-induced phosphorylation of cAMP-activated protein kinase (PKA) and Hormone Sensitive Lipase (HSL), thus eventually modulating lipolysis [22]. In Siberian hamsters, TLQP-21 strongly affects metabolic function [41]. VGF over-expression in hypothalamic area or i.c.v. TLQP-21 injection increase energy expenditure and reduce body weight gain [23]. Moreover, systemic administration of TLQP-21 in lean hamsters exposed to short, but not long photoperiods, stimulates energy expenditure and decreases food intake [42]. This could be explained with the modulation of VGF expression in the hypothalamus during photoperiod variation [43,44]. 

### 2.4. TLQP-21: Extraendocrine Activities 

It is challenging to exactly define TLQP-21 actions, since the source of VGF-derived peptides have not been completely ascertained. In this section we will address TLQP-21 targets that are not directly involved with endocrine functions.

#### 2.4.1. Neuroprotection 

VGF expression has been implicated in different models of neuroprotection and antidepressant action [45]. Only few studies, instead, have investigated directly the role of TLQP-21.

TLQP-21 increased cell survival and decreased apoptotic death in primary cell culture of cerebellar granule cells [46]. The cerebellar granule cells were cultured in a depolarization milieu (extracellular 25 mM potassium concentration) that stimulates secretion of different endogenous growth factors including VGF-derived peptides. Lowering the extracellular potassium to 5 mM, a level that stops the secretion of autocrine survival factors, induced apoptotic cell death. VGF peptides are released by cell depolarization, suggesting autocrine and/or paracrine actions. Among different growth factors secreted by cell cultures, TLQP-21 has been clearly demonstrated to be involved in cell survival through phosphorylation of mitogen-activated protein kinase (MAPK) and a Ca^2+^-dependent mechanisms [46]. 

Concerning VGF and its derived peptides, recent studies indicated a strong role in cerebellar development and function [47,48]. VGF expression in rat brain was extensively described [49]. Retina and visual system express high level of VGF during synaptogenesis [50,51]. These studies prompted further investigations on the role of some VGF-derived peptides on retinal cells. In vitro, co-cultures of stem cells derived from different tissues with retinal ganglion cells, showed paracrine secretion of protective anti-apoptotic and neurogenic factors on retinal cells [52]. Among the trophic factor identified, TLQP-21 significantly increased cell survival (1–10 µM), without displaying effects on neuritogenic axon regeneration. 

Finally, recent evidence shows that TLQP-21 could be considered a biomarker for neurodegenerative diseases, such as Amyotrophic Lateral Sclerosis (ALS) [53]. Indeed, Brancia and coworkers demonstrated that TLQP peptides expression decreases both in vitro, in stressed NSC-34 cells and untreated fibroblast cultures from ALS patients, as well as in vivo, in motor neurons of Superoxide Dismutase-1 (SOD1) mice at the earliest, pre-symptomatic stage of the pathology, before the onset of significant muscle weakness took place. This reduction is similar to that observed in the plasma of ALS patients at the early clinical stages. These plasma alterations may reflect early changes occurring in motor neurons, suggesting that TLQP peptides could be investigated as indicators for early diagnosis of ALS. The decrease of TLQP-21 correlates with the rise of oxidative stress and the exogenous administration of the peptide to stressed NSC-34 cells protects the cells from death [53]. 

#### 2.4.2. Stress Responses, Inflammation and Nociception 

Among the extraendocrine actions, a link between VGF-derived peptides and mood disorders has been proposed. Recent studies, in fact, have highlighted a possible role for VGF-derived peptides in depression and neuropsychiatric disorders, for which social stress is a known key-player [54,55,56,57]. In particular, a role for TLQP-21 was defined in stress responses, specifically in acute restraint stress (RS) and chronic subordination stress (CSS). Indeed, Razzoli and collaborators, investigating the action of VGF-derived peptide in centrally mediated stress responses found that, upon central administration, TLQP-21 was able to increase serum epinephrine (E) and decrease norepinephrine (NE) levels after acute restrain [58]. The variation of plasmatic E/NE balance in favor of a major release of E is generally considered a marker of stress [59], thus supporting the idea of a TLQP-21 involvement in stress responses. Moreover, when TLQP-21 was chronically administrated, it worsened the behavioral syndrome induced by chronic social stress, exacerbating the social avoidance exhibited by stressed mice [58]. 

These interesting results have brought to surface two important aspects. First, the results presented by Bartolomucci and coworkers appear somehow in contrast with other observations showing that the lack of VGF gene, and therefore assuming the absence of VGF-derived peptides, increased depression-related behavioral changes. In the same direction is the observation that the acute treatment with some of VGF peptides (i.e., TLQP-62 and AQEE-30) is associated to anti-depressant like effects in rats and mice [56]. However, opposite effects exerted by other VGF-derived peptides have also been described: for instance, TLQP-21 in CNS shows a catabolic activity, while NERP-2 modulates metabolic functions and adiposity in opposite directions [60,61]. Second, when investigating TLQP-21 mechanism of actions the route of administration needs to be carefully considered. Indeed, TLQP-21 infusion turned out to exert a differential effect on circulating catecholamine concentration depending on the route of infusion [20,40,58]. On the other hand, E levels seem not to be altered by peripheral infusion in obese mice [40].

The importance of the route of administration in TLQP-21 actions is underlined also by a study showing that TLQP-21 could regulate pain [62,63]. By means of behavioral formalin tests, it has been demonstrated that TLQP-21 was able to induce hyperalgesia when injected peripherally at high doses, but not at low doses. Furthermore, when locally administered, TLQP-21 effect was detected only during the second phase, characteristic of the s.c. formalin injection, in the formalin test. It has been postulated that TLQP-21 could be secreted in response to nociceptive stimuli together with substance P, a well-known modulator of nociceptive pain, since they co-localize in the secretory granules of dorsal root ganglion (DRG) neurons [62]. In the attempt to identify the mechanisms of action of TLQP-21 in regulating pain, it has been proposed that TLQP-21 may participate to inflammation through a prostaglandin-mediated mechanism. In fact, VGF peptide turned out to stimulate COX activity and the production of prostaglandin E2. Thus, it appears that TLQP-21 is able to induce an analgesic effect on formalin-induced pain if injected via i.c.v., but it causes hyperalgesia when injected peripherally, thus suggesting that TLQP-21 could have opposite effects depending on the neuronal level on which it is acting [62].

Another interesting study provided evidence showing that TLQP-21 participates in mediating spinal neuroplasticity after inflammation and nerve injury [64]. Fairbanks and colleagues demonstrated that the exogenous application of TLQP-21 induced dose-dependent thermal and tactile hypersensitivity. The activation of a p38 MAPK and COX and lipoxygenase (LOX) pathways resulted to be involved [64].

### 2.5. TLQP-21: Mechanism of Action 

Considering the plethora of actions displayed by TLQP-21, the need of identifying a specific receptor became compelling to focus on novel targets for pharmacological intervention. The first efforts made in this direction have shown that TLQP-21 was able to induce a transient intracellular Ca^2+^ surge, in a dose dependent manner, in macrophages and primary microglia [63], GH3 pituitary cells [25], primary culture of cerebellar granule cells [46] Chinese Hamster Ovary cells (CHO-K1) [65,66], N9 microglial cells [67], and RAW264.7 macrophages [68], indicating the existence of a specific binding site. The specificity of the binding was supported by the following evidences: (i) TLQP-21 resulted unable to induce free Ca^2+^ mobilization in other cell lines, including N38, N41, and N42 cells [64,65,66]; (ii) the effect on Ca^2+^ release is specific for TLQP-21 and was not replicated with other VGF-derivatives or with LRPS-21, a peptide that contains the same 21 amino acid residues of TLQP-21, but arranged in a scrambled order to avoid any homology with other known proteins [64,65,66,67]; (iii) the binding of [125]-YATL-23, a radiolabeled TLQP-21 peptide, on crude membranes obtained from CHO-K1 cells was specific and not inhibited by LRPS-21 [67]. The hypothesis of the presence of a single class of binding sites on CHO-K1 cells was corroborated also by data obtained using the atomic force microscopy (AFM) in force mapping mode with a cantilever functionalized with TLQP-21 [68]. By quantifying the interaction forces between the ligand and the binding site, and mapping the topological distribution of the binding sites on the cell surface, AFM experiments allowed to measure the attraction force between binding sites on CHO-K1 membranes and TLQP-21, showing features typical of classical ligand–receptor interaction (about 39 ± 7 pN). 

Further studies performed in CHO-K1 cells identified the complement C3a receptor-1 (C3aR1) as the target site of TLQP-21 [69]. C3aR1 is a member of the G protein-coupled receptor (GPCR) superfamily, characterized by the classic seven transmembrane domains motif and a large second extracellular loop [70]. C3aR1 is primarily known for its involvement in regulation of the innate immune response and prompt surveillance for defense from pathogens. Indeed, it plays a key role in keeping the complement cascade alert, engaging the amplification of the immune response and helping to coordinate downstream responses [71]. However, recent researches provided evidence about a role also in other conditions, ranging from cancer [72] to neurogenesis [73], insulin resistance [74], pituitary hormone release [75], and lipolysis [65,76]. Concerning the lipolytic process, data obtained in rodents suggest a link between C3aR1 and TLQP-21. Interestingly, obese mice display low levels of TLQP-21 and increased expression of C3aR1, whereas C3aR1 knock-out (KO) mice are transiently resistant to diet-induced obesity and are protected against insulin resistance induced by a high-fat diet [76]. 

In CHO-K1 cells the binding of TLQP-21 on C3a1R stimulated an intracellular Ca^2+^ increase only after ATP priming: this process seemed not to be coupled to cAMP or inositol phosphate accumulation, but likely mediated by a G_o_ protein, since it could be blocked by pertussis toxin (PTX) [69]. However, these data are not completely in agreement with our data supporting a G_q_ protein-mediated mechanism, by which the rapid Ca^2+^ mobilization induced by TLQP-21 is explained by the release of Ca^2+^ from intracellular stores, in particular the ER, whereas the mitochondria did not appear involved [66]. Actually, the pretreatment of CHO-K1 cells with TG, which inhibits the Ca^2+^-ATP-ase pump responsible for Ca^2+^ sequestering in the ER and depletes the store by irreversibly preventing its refilling, reduced the Ca^2+^ mobilization stimulated by TLQP-21 [66]. Accordingly, TLQP-21 administration to CHO cells quickly activates Phospholipase C (PLC)-β, producing the second messengers Inositol Trisphosphate (IP3) and Diacylglycerol (DAG). DAG activates Protein kinase C (PKC), that in turn, stimulates the extracellular signal-regulated kinase (ERK) 1/2 phosphorylation. On the other hand, by activating the IP3 receptor on ER surface, IP3 enhances Ca^2+^ secretion from ER store and the subsequent Ca^2+^ entry from the outside of the cell. The Ca^2+^ influx could depend on STIM-CRAC-Orai1 interaction, as demonstrated by data showing that SKF-96365 and YM-58483, two inhibitors of STIM and CRAC-Orai1, respectively, both inhibit TLQP-21-stimulated Ca^2+^ release [68]. In agreement with these data, also in 3T3L-1 cells the pretreatment with EGTA, a Ca^2+^-chelator, prevented TLQP-21-induced lipolysis mediated by HSL activation via Ca^2+^ entry from the extracellular space [76]. However, the role of CRAC-Orai1 is still debated since only the use of SKF-96365, a pan-specific transient receptor potential channel (TRPC) inhibitor, and not that of AncoA4, a specific inhibitor of Store-Operated Calcium (SOC) release mediated by CRAC-Orai1, blocks TLQP-21-induced increase in intracellular Ca^2+^ [77]. Experiments performed in N9 cells show that following extracellular Ca^2+^ entry stimulated by TLQP-21, the activation of outward K^+^-Ca^2+^-dependent currents, mainly the KCa 3.1 current, provides a membrane iperpolarizing effect which offsets the depolarizing action of Ca^2+^ elevation and indirectly maintains the driving force for optimal Ca^2+^ increase in the cytosol [67]. Figure 2 summarizes TLQP-21 intracellular transduction mechanism in CHO-K1 cells.

Although C3aR1 has been proposed as the putative receptor for TLQP-21 [71], some data argue against this hypothesis. In RAW264.7 cells that express C3aR1, the mobilization of intracellular Ca^2+^ can be induced by stimulation with C3a, the natural ligand of C3aR1, and C3a_(70–77)_ peptide, an active fragment of C3a, as well as with TLQP-21 and JMV5656, a novel short analogue of TLQP-21 (TLQP-21^9-21^) [68]. However, in the same cells, the existence of two different receptors could be suggested by the following results: 1) repeated administration of JMV5656 induced a progressive reduction of Ca^2+^ release elicited by subsequent stimulations with JMV5656, a likely index of homologous receptor desensitization; 2) repeated C3a_(70−77)_ stimulations, instead, consistently elicited full intracellular Ca^2+^ responses; 3) the use of siRNAs against C3aR decreased intracellular Ca^2+^ mobilization stimulated by C3a but not JMV5656 [68]. Although these experiments were performed in only one cell line, and we cannot completely rule out that in other cell lines the outcome could be different to that of the RAW264.7 cells, they suggest that more than one specific receptor mediates C3a and TLQP-21 effects. 

RAW264.7 express also significant amounts of mRNA for gC1qR [68], the receptor for globular heads of the complement component C1q, which has been also proposed capable to bind TLQP-21 in macrophages [63]. In these cells, it cannot be excluded that JMV5656 could activate both C3aR and gC1qR.

## 3. Conclusions 

The VGF-derived peptide TLQP-21 has been extensively investigated because of its various actions and properties in different experimental settings. Some of its properties have been characterized, but others should be object of further studies in future, with the aim to better understand its specific role. In this context, the knowledge of the potential activity of TLQP-21 in preventing or reducing motor neuron death should be a stimulus to engage further studies aimed to address the possible relevance of TLQP peptides for new therapeutic approaches in neurodegenerative diseases. On the other hand, the possible role of TLQP-21 and other VGF-derived peptides as potential biomarkers of the early phases of disease onset and/or progression in ALS is also worth of further investigations. 

Finally, several tiles of the complex mosaic of TLQP-21 mechanism of action have been sketched out, but many others have yet to be clarified. The identification of the real specific receptor(s) is a starting point to boost the pharmacological research for the development of novel agonist and antagonist of TLQP-21. 

## Figures and Tables

**Figure 1 ijms-21-00130-f001:**
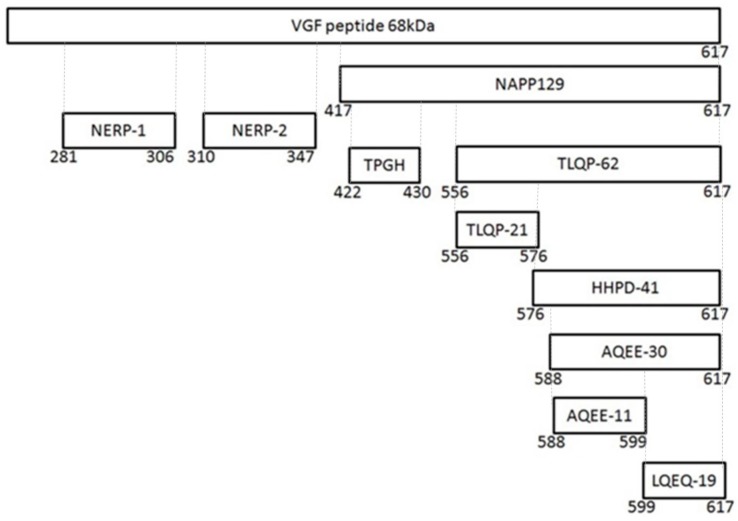
Schematic representation of VGF and of VGF-derived peptides.

**Figure 2 ijms-21-00130-f002:**
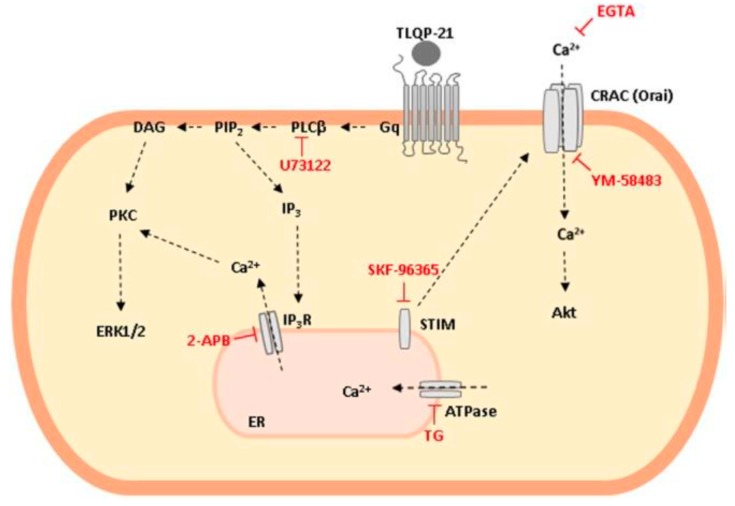
Schematic representation of TLQP-21 intracellular transduction mechanism in Chinese Hamster Ovary (CHO-K1) cells. TLQP-21, by binding a G protein coupled receptor (GPCR), activates Phospholipase C (PLC)-β that in turn produces Diacylglycerol (DAG) and Inositol Trisphosphate (IP3) as second messengers. These molecules activate Protein kinase C (PKC), stimulate extracellular signal-regulated kinase (ERK)1/2 phosphorylation and induce intracellular Ca^2+^ release from the endoplasmic reticulum (ER) with the subsequent Ca^2+^ entry from outside the cell, mediated by the stromal interaction molecules (STIM) and calcium release-activated calcium channel (CRAC)-Orai1 interaction. Phosphorylation of AKT is probably a result of the increase in cellular Ca^2+^ concentrations.

**Table 1 ijms-21-00130-t001:** VGF and derived peptides.

Name	Fragment	Espression	Potential Role	Reference
VGF	0–617	Hypothalamus Hippocampus Amygdala Thalamus Cerebral Cortex Pituitary Adrenal Medulla Gut Pancreas	Energy balance Reproduction Momory Learning Depression	Lewis J.E.; et al.; 2015
NERP-1	281–306	Hypothalamus Thyroid Gastric Antrum	Inhibitory modulators of vasopressin relaease	Toshinai K.; et al.; 2009
NERP-2	310–347	Hypothalamus Thyroid Pancreas Gastric antrum	Inhibitory modulators of vasopressin relaease Stimulator offeeding behavior	Toshinai K.; et al.; 2009
			Enhancer of glucose-stimulated insulinsecretion	Moin A.S.; 2012
			Increased gastric acid secretion and gastric emptying	Namkoong C.;et al.; 2017
TLQP-62	556–617	Hypothalamus Hippocampus	Enhanced synaptic activity	Alder J.; et al.; 2003
			Effects on spontaneous excitability of superficial dorsal horn neurons	Moss A.; 2008
			Antidepressant effects	Hunsberger J.G.; et al.; 2007
			Spinal plasticity	Skorput A.G.J.; et al.; 2018;.
			Long-term memory formation	Lin W.-J. et al.; 2015
AQUEE-30	588–617	Pituitary	In vitro neuroprotective effects	Noda Y.; et al.; 2019
			Enhanced synaptic activity	Alder J.; et al.; 2003
			Antidepressant effects	Humsberger J.G.; et al.; 2007
			Pro-nociceptive and hyperalgesic functions	Riedl M.S.; et al.; 2009
			Thermal hyperalgesia	Riedl M.S.; et al.; 2009
LQEQ-19	599–617	Thalamus cerebral cortex	In vitro neuroprotective effects	Noda Y.; et al.; 2019
			Pro-nociceptive and hyperalgestc functions	Riedl M.S.; 2009
			Thermal hyperalgesia	Riedl M.S.; 2009

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
