# Peer review of "TLQP-21, A VGF-Derived Peptide Endowed of Endocrine and Extraendocrine Properties: Focus on In Vitro Calcium Signaling"

_ijms, 2019, doi:10.3390/ijms21010130_

Round 1
Reviewer 1 Report
This seems to be a good paper. My comments:
Figure 1 should uploaded in better resolution A figure would be nice on VGF and VGF derived peptides and their role in CNS and PNS pathways A table may also would be nice, which could summarize the expression and the potential roles of VGF derived peptides
Author Response
Thank you for your suggestions; we have now introduced Table 1 indicating the role of several VGF-derived peptides.
Reviewer 2 Report
In this review Bresciani et al, focused their attention on the physiological role and mechanism of action of TLQP-21 peptide generated by the VGF neuropeptide by evidencing the crucial of intracellular Ca2+ as the intracellular second messanger. The wide range of physiological actions (endocrine, extraendocrine, metabolic…) have been well described by this review but from my point of view some minor aspects would be considered to clarify some details just to make it more fluent to read:
The “depletion of intracellular Ca stores is sensed by Stromal Interaction Molecules (STIM) which interact with ...for store-operated Ca entry (SOCE) currents”, cited here for the first time (page 2 line 53). Different mechanisms responsible for the maintenance of intracellular calcium concentration have been cited in the text and easy to find in the very clear schema of fig.2 but among all ones cited, it is not so evident that the SOCE current is identifiable as CRAC-Orai1 channels that interact with STIM ones. I think that the schema would be implemented with all elements cited in the text to complete it or alternatively, the text would be completed with a short sentence to better explain them.
PAGE 9 line 353-363. The paragraph is difficult to read and to focus. The existence of two receptors, cited peptides and analogues (JMV5656 as TLQP-21 derivative (Rivolta, 2017)) and the functional link are not so evident, please rephrase more clearly.
Generally, according to the title we expect an overview on the calcium signaling role in different physiological functions of the peptide, but actually, although the review is well written and different mechanisms are considered in detail in a wide range of physiological roles, from my point of view, the calcium signaling is focused only on the restricted cell population. This leaves the reader with the possibility to clarify the mechanisms with future studies, then I think that the title it should be slightly adapted to the real overview of the paper. Alternatively, the authors may further elaborate what has been already described in calcium signaling mechanisms in different cells and physiological role, considering that TLPQ1-21 is also able to induce different effects (analgesic or hyperalgesic) depending on the neuronal level of its action.
Author Response
We thank the reviewer for the suggestions; the text has been amended accordingly. Now we use the abbreviation CRAC-Orai1 to armonize the text with Figure 2.
Consistently, we have rectified all the acronyms in the text, in the key words, in the legend of the Figure 2, in Figure 2 and in the abbreviation list.
Reviewer 3 Report
Bresciani et al., in the manuscript titled “TLQP-21 calcium signaling in endocrine and extra-endocrine activities,” review TLQP-21, one of the cleavage products derived from VGF a neuropeptide. They overviewed the cellular calcium homeostasis, Ca2+ influx and efflux, and ER Ca2+ release and summarized various cleavage peptides of VGF in the introduction. A variety of TLQP-21 actions related to endocrine, reproduction, neuroprotection, and inflammation, etc., and TLQP-21 targets, Ca2+ surge signaling, and triggering of ER Ca2+ release are the main body of the review.
The review was analytical in writing and the abstract is adequate. The streaming of the recent progress is well organized, being a quite attentive contribution to the VGF research field.
Minor issue:
Line 25 or in other multiple places: highlighting recent findings on his mechanism of action and his potential in different
Comment: Neutral word sounds natural. (Why is it not her?)
2. Line 84: endocrine cells [9]. In the CNS, VGF immunoreactivity is particularly evident in the olfactory
Comment: looking for a brief explanation about VGF immunoreactivity, but is missing in the following.
3. If a brief overview of the distributions and functions of other VGF derived peptides in the introduction is included, not only the list, the review would provide a better viewpoint of the topic.
Author Response
Rev 2 Comments and Suggestions for Authors
The “depletion of intracellular Ca stores is sensed by Stromal Interaction Molecules (STIM) which interact with ...for store-operated Ca entry (SOCE) currents”, cited here for the first time (page 2 line 53). Different mechanisms responsible for the maintenance of intracellular calcium concentration have been cited in the text and easy to find in the very clear schema of fig.2 but among all ones cited, it is not so evident that the SOCE current is identifiable as CRAC-Orai1 channels that interact with STIM ones. I think that the schema would be implemented with all elements cited in the text to complete it or alternatively, the text would be completed with a short sentence to better explain them.
We thank the reviewer for the suggestions; the text has been amended accordingly. Now we use the abbreviation CRAC-Orai1 to armonize the text with Figure 2.
Consistently, we have rectified all the acronyms in the text, in the key words, in the legend of the Figure 2, in Figure 2 and in the abbreviation list.
PAGE 9 line 353-363. The paragraph is difficult to read and to focus. The existence of two receptors, cited peptides and analogues (JMV5656 as TLQP-21 derivative (Rivolta, 2017)) and the functional link are not so evident, please rephrase more clearly.
As suggested by the reviewer, we enlarged and amended the paragraph, hoping that now it is more clear for the reader (lines 356-372).
Generally, according to the title we expect an overview on the calcium signaling role in different physiological functions of the peptide, but actually, although the review is well written and different mechanisms are considered in detail in a wide range of physiological roles, from my point of view, the calcium signaling is focused only on the restricted cell population. This leaves the reader with the possibility to clarify the mechanisms with future studies, then I think that the title it should be slightly adapted to the real overview of the paper. Alternatively, the authors may further elaborate what has been already described in calcium signaling mechanisms in different cells and physiological role, considering that TLPQ1-21 is also able to induce different effects (analgesic or hyperalgesic) depending on the neuronal level of its action.
We agree with the reviewer that the title (TLQP-21 calcium signalling in endocrine and extraendocrine activities) is a bit too generic and that it could not completely match with the topics discussed. Following this suggestion, we have changed the title:
TLQP-21, a VGF-derived peptide endowed of endocrine and extraendocrine properties: focus on in vitro calcium signalling
REV 3 Minor issue:
Line 25 or in other multiple places: highlighting recent findings on his mechanism of action and his potential in different
Comment: Neutral word sounds natural. (Why is it not her?)
Sorry for typos; now TLQP-21 has been indeed considered as neutral name.
Line 84: endocrine cells [9]. In the CNS, VGF immunoreactivity is particularly evident in the olfactory
Comment: looking for a brief explanation about VGF immunoreactivity, but is missing in the following.
We have now introduced a paragraph to explain the possible role of VGF expression in these neuronal and peripheral systems.
If a brief overview of the distributions and functions of other VGF derived peptides in the introduction is included, not only the list, the review would provide a better viewpoint of the topic.
We introduced Table 1 to show the purported activities of VGF-derived peptides mentioned in the text.